# External Validation of the Graded Prognostic Assessment in Patients with Brain Metastases from Small Cell Lung Cancer

**Carsten Nieder [1,2,\*], Ilinca Popp [3], Mandy Hintz [3] and Anca Ligia Grosu [3]**

[1] Department of Oncology and Palliative Medicine, Nordland Hospital Trust, 8092 Bodø, Norway

[2] Department of Clinical Medicine, Faculty of Health Sciences, UiT—The Arctic University of Norway, 9037 Tromsø, Norway

[3] Department of Radiation Oncology, University Hospital Freiburg, 79106 Freiburg, Germany

[\*] Correspondence: carsten.nieder@nlsh.no; Tel.: +47-755-78-490

**Abstract:** Background: Recently, graded prognostic assessment (GPA) for small cell lung cancer (SCLC) patients with brain metastases has been developed. This includes age, performance status, number of brain metastases and presence of extracranial metastases. The aim of the present study was to validate this four-tiered prognostic score in a European cohort of patients. Methods: The retrospective validation study included 180 patients from two centers in Germany and Norway. Results: Median survival from radiological diagnosis of brain metastases was 7 months. The GPA point sum as continuous variable (0–4 points) was significantly associated with survival ($p < 0.001$). However, no significant survival difference was observed between patients in the two strata with better survival (3.5–4 and 2.5–3 points, respectively). Long-term survival in the poor prognosis group (0–1 points) was better than expected. Conclusion: This study supports the prognostic impact of all four parameters contributing to the GPA. The original way of grouping the parameters and breaking the final strata did not give optimal results in this cohort. Therefore, additional validation databases from different countries should be created and evaluated.

**Keywords:** radiation therapy; stereotactic radiotherapy; brain metastases; prognostic factors; small cell lung cancer

## 1. Introduction

Prognostic models predicting survival after treatment of brain metastases have recently undergone substantial refinement [1]. Additional nomograms and scores have been developed and validated. Both diagnosis-specific and site-agnostic scores are available to clinicians who would like to avoid mismatch between treatment intensity and achievable outcome [2–6]. As part of this effort, Sperduto et al. have updated the graded prognostic assessment (GPA) for lung cancer patients [7]. For the first time, a separate GPA for small cell lung cancer (SCLC) has been introduced. Their multi-institutional study included 570 patients treated between 2015 and 2020 (USA, Canada, Japan). Commonly ($n = 314$), whole-brain radiotherapy (WBRT) was administered, while 108 patients received stereotactic radiosurgery (SRS). In 30 cases, surgical resection was a component of care. The latter approach can be considered in selected patients with single brain metastasis [8]. Most patients in the GPA study also had systemic chemotherapy (before local brain-directed treatment: 331, afterwards: 170). Significant prognostic factors for survival were age, Karnofsky performance status (KPS), extracranial metastases and the number of brain metastases (Table 1). Gender, race and ethnicity were not significant.

Median survival from diagnosis of brain metastases was 10 months (9 after WBRT). The median values for GPA scores of 0–1.0, 1.5–2.0, 2.5–3.0 and 3.5–4.0 were 4, 8, 13 and 23 months, respectively. Each group contained patients who were alive after 24 months. The purpose of the present study was to analyze the validity of the SCLC GPA in patients

treated in two European institutions, i.e., a region where different treatment strategies may be employed.

**Table 1.** Calculation of the point sum that determines prognostic group. Worst prognosis (0 points): Karnofsky performance status <60 and >7 brain metastases and presence of extracranial metastases and age $\geq$ 75 years. Worst prognostic class: 0–1 points (other classes: 1.5–2 points, 2.5–3 points, 3.5–4 points).

| Baseline Parameter | 0.5 Points | 1 Point | 1.5 Points | 2 Points |
|---|---|---|---|---|
| Karnofsky performance status | 70 | 80 | 90 | 100 |
| Number of brain metastases | 4–7 | 1–3 | | |
| Extracranial metastases | None | | | |
| Age (years) | < 75 | | | |

## 2. Materials and Methods

We followed the design and methods that were utilized in comparable validation studies by our group [9,10]. Patients with brain metastases from SCLC were identified from the institutional review board-approved databases at Nordland Hospital Trust Bodø, Norway, and University Hospital Freiburg, City, Germany. Inclusion criteria: 2006–2021, parenchymal brain metastases from histologically verified extracranial primary SCLC managed with WBRT, SRS, surgery or upfront chemotherapy followed by salvage radiation. Both, completed and interrupted treatment courses were included according to the intention-to-treat principle. Some patients had received previous prophylactic brain irradiation (PCI). In this real-world cohort, treatment sequence and radiotherapy prescription were individualized, and so was further treatment for new or recurrent brain metastases. Systemic treatment was continued or initiated as judged appropriate by the multidisciplinary lung cancer tumor boards at the study sites. The prevailing dose-fractionation regimen was 30 Gy in 10 fractions. A minority of patients received 20 Gy in 5 fractions or WBRT with additional boost, typically simultaneously integrated. After initial PCI, SRS or fractionated focal radiotherapy was preferred. However, selected patients were managed with a second course of WBRT.

Extracranial staging consisted of computed tomography (CT). If clinically relevant, further modalities were added to clarify CT findings, e.g., isotope bone scan, ultrasound, positron emission tomography (PET). The number of brain metastases was derived from magnetic resonance imaging (MRI) reports. Overall survival (time to death) from radiological diagnosis of brain metastases was calculated employing the Kaplan–Meier method, and different groups were compared using the log-rank test (SPSS 27, IBM Corp., Armonk, NY, USA). Eight of 180 patients were censored after median 16.5 months of follow-up. Date of death was known in all other patients. For continuous variables, such as age and GPA, univariate Cox regression was employed. The GPA score was calculated as proposed by Sperduto et al [7]. (Table 1).

## 3. Results

The study included 180 patients, largely managed with WBRT (7% resection or SRS). The median Karnofsky performance status (KPS) was 70, the median age 64 years. Fifty-six percent had four or more brain metastases, and 71% extracranial metastases. Table 2 provides further baseline characteristics.

KPS, age and presence of extracranial metastases predicted survival, as also shown in Table 2. Number of brain metastases reached $p < 0.001$ if analyzed as continuous variable (Cox regression), whereas the grouping employed in the GPA failed to achieve statistical significance. None of the other parameters, which are not part of the GPA, was statistically significant.

**Table 2.** Patient characteristics (*n* = 180).

| Baseline Parameter | Number | Percent | Significance (OS), *p*-Value and Hazard Ratio (HR) |
|---|---|---|---|
| Female sex | 85 | 47 | |
| Male sex | 95 | 53 | 0.81, HR 1.1 |
| KPS ≤ 60 | 47 | 26 | |
| KPS 70 | 49 | 27 | |
| KPS 80 | 33 | 18 | |
| KPS 90 | 31 | 17 | |
| KPS 100 | 20 | 11 | <0.001, HR 0.92 |
| No extracranial metastases | 52 | 29 | |
| Extracranial metastases | 128 | 71 | 0.04, HR 2.8 |
| Controlled primary tumor | 79 | 44 | |
| Uncontrolled primary tumor | 101 | 56 | 0.16, HR 1.7 |
| 1–3 brain metastases | 79 | 44 | |
| 4–7 brain metastases | 40 | 22 | |
| ≥8 brain metastases | 61 | 34 | 0.25, HR 1.6 |
| Synchronous brain metastases | 99 | 55 | |
| Metachronous brain metastases | 81 | 45 | 0.41, HR 0.8 |
| Symptomatic brain metastases | 90 | 50 | |
| Staging-detected brain metastases | 90 | 50 | 0.44, HR 0.9 |
| Largest lesion < 2 cm diameter | 104 | 58 | |
| Largest lesion 2–3 cm diameter | 43 | 24 | |
| Largest lesion > 3 cm diameter | 33 | 18 | 0.22, HR 1.5 |
| Neurosurgical resection | 5 | 3 | |
| Primary SRS/focal radiotherapy | 7 | 4 | |
| Chemotherapy naïve | 52 | 29 | |
| Chemotherapy before local therapy of brain metastases | 128 | 71 | |
| Median age, range, mean and SD (years) | 64, 35–88 | 64, 10 | 0.005 *, HR 1.04 |
| Age younger than 75 years | 154 | 86 | |
| Age 75 years or older | 26 | 14 | <0.001, HR 2.9 |

OS: overall survival, KPS: Karnofsky performance status, SRS: stereotactic radiosurgery. * Cox regression, continuous variable (other parameters: log-rank test; only patient- and disease-related parameters were assessed with univariate tests, while treatment-related parameters were not).

The GPA point sum as continuous variable (0–4 points) was significantly associated with survival (Cox regression *p* < 0.001). Table 3 displays the number of patients per GPA stratum and group, and their survival outcomes.

The Kaplan-Meier curves are shown in Figure 1 (detailed) and Figure 2 (grouped). No significant survival difference was observed between patients with 3.5–4 and 2.5–3 points, respectively. Long-term survival in the poor prognosis group (0–1 points) was better than expected, as a result of the patients with 1 point, who had distinctly better survival than those with 0–0.5 points.

**Table 3.** Graded prognostic assessment (GPA) and actuarial survival (*n* = 180, Kaplan-Meier analysis).

| GPA | Number of Patients | Median Survival (mo) | 1-Year Rate (%) | 2-Year Rate (%) |
|---|---|---|---|---|
| 0 | 7 | 1.3 | 0 | 0 |
| 0.5 | 12 | 1.1 | 0 | 0 |
| 1 | 28 | 4.0 | 19 | 12 |
| 1.5 | 32 | 6.0 | 16 | 3 |
| 2 | 35 | 8.0 | 18 | 3 |
| 2.5 | 31 | 9.7 | 33 | 7 |
| 3 | 18 | 13.0 | 61 | 28 |
| 3.5 | 13 | 13.0 | 54 | 15 |
| 4 | 4 | 8.0 | 25 | 25 |
| Group 1 | 47 | 3.0 | 11 | 7 |
| Group 2 | 67 | 6.9 | 21 | 3 |
| Group 3 | 49 | 11.0 | 46 | 15 |
| Group 4 | 17 | 11.0 | 47 | 18 |
| Combined | 180 | 7.0 | 28 | 9 |

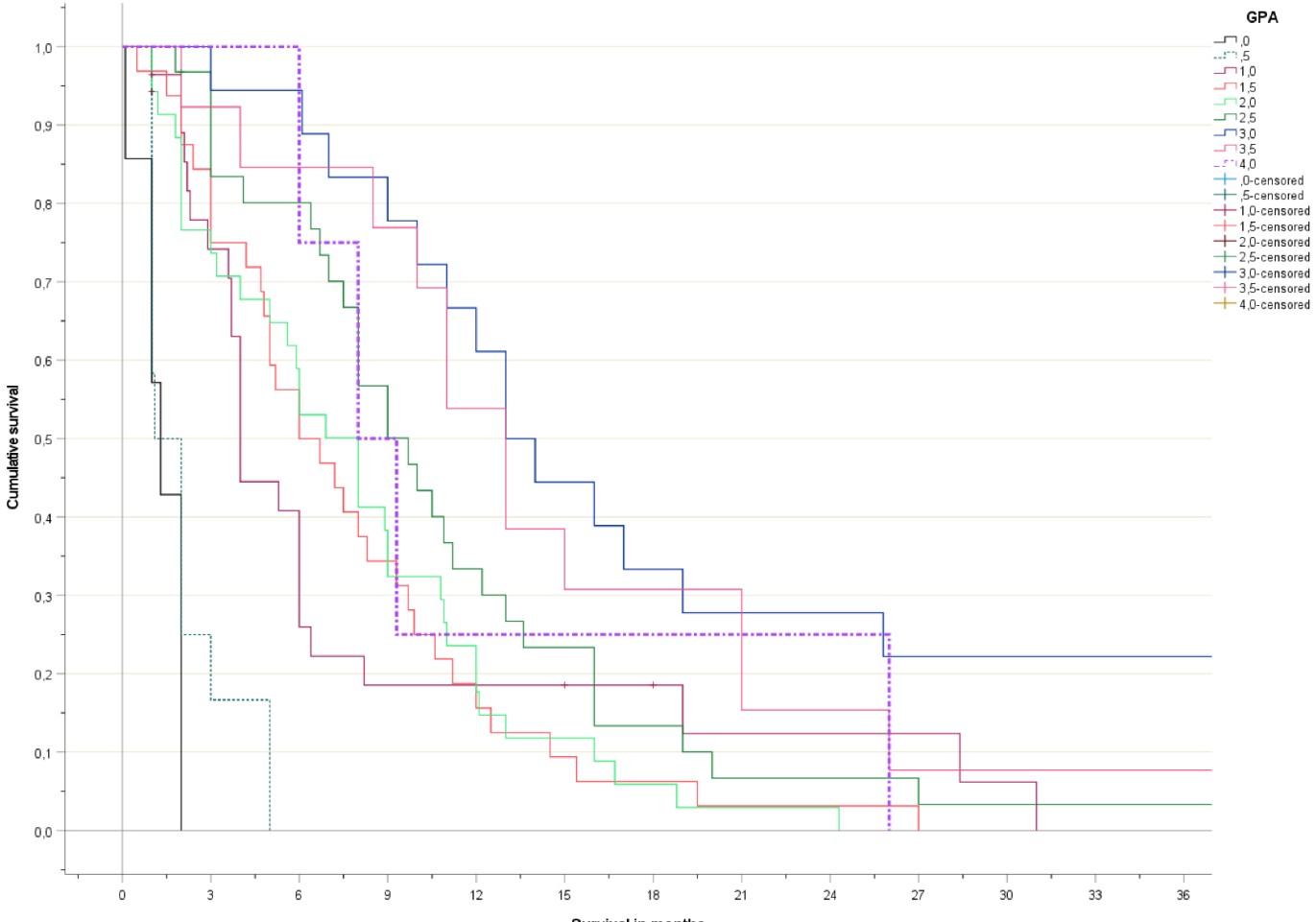

**Figure 1.** Actuarial overall survival (Kaplan-Meier curves, log-rank test pooled over all strata *p* < 0.001).

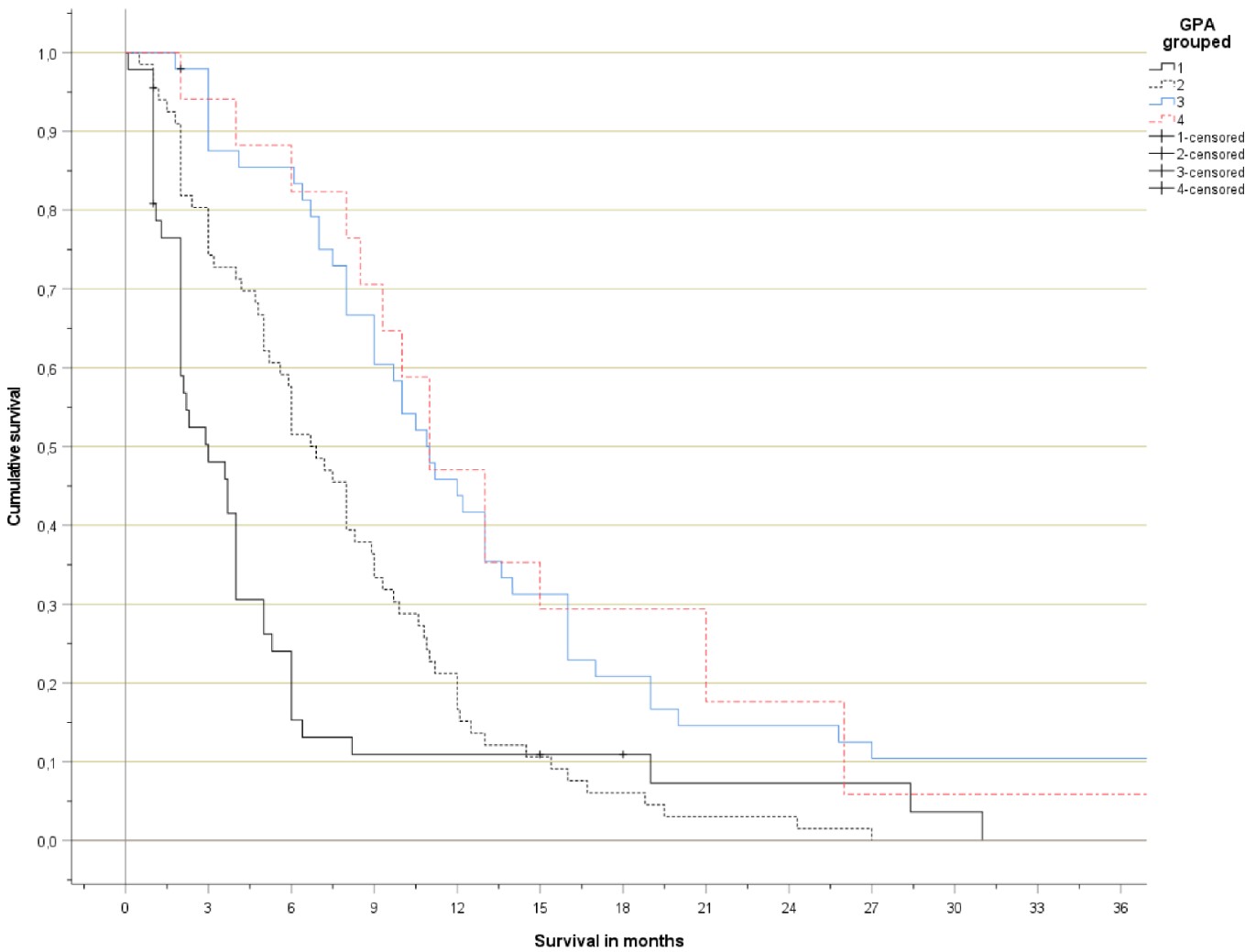

**Figure 2.** Actuarial overall survival (Kaplan-Meier curves, log-rank test pooled over all strata $p < 0.001$; $p < 0.05$ for all pairwise comparisons except group 3 versus 4).

## 4. Discussion

The present study aimed at validation of the SCLC GPA in a different geographical region, where management approaches are not necessarily identical to those utilized in the cohort evaluated by Sperduto et al. [7]. Indeed, we found different rates of SRS/surgery utilization (4 and 3% (present) versus 19 and 5% (Sperduto et al.), respectively). Symptomatic brain metastases were common in our study (50%) and 42% had index lesions $\geq$ 2 cm in maximum diameter (details not reported by Sperduto et al.). Fewer patients had 1–3 brain metastases (44% in the present study versus 52%). More patients had KPS $\leq$ 60 (26% versus 14% in the Sperduto et al. study). The proportion of patients with extracranial metastases was higher (71% versus 62% in the Sperduto et al. study). Other notable differences include cohort size (180 and 570 patients, respectively) and treatment period (2006–2021 and 2015–2020, respectively). In summary, the present cohort was characterized by more advanced disease, poorer KPS and thus, impaired prognosis. As one might expect, median survival was shorter in our study (7 months) compared to Sperduto et al. (10 months). The survival difference between the studies was seen in all four prognostic groups (median 3 versus 4 months, 6.9 versus 8 months, 11 versus 13 months, and 11 versus 23 months, respectively). The most striking difference occurred in the best prognostic group. Possible explanations include different numbers of patients who underwent MRI screening and therefore had less advanced intracranial disease, or differences in SRS and chemotherapy utilization. As reported by Putora et al., decision-making and management approaches

for SCLC brain metastases are heterogeneous [11], and also PCI, which reduces the risk of brain metastasis development, is not routinely utilized in all patients [12,13].

We confirmed the GPA point sum and most groups as significant prognostic factors. However, similar survival curves were found for the two groups with better prognosis. In principle, this finding might just be a consequence of the small group size (inclusion time period was extended to mitigate this as far as possible) and might vanish in a larger study. On the other hand, we were unable to confirm the appropriateness of the GPA brain metastases stratification (1–3, 4–7, ≥8). If only three of four prognostic factors (age, KPS, extracranial metastases) actually are associated with survival, the resulting score misses the discriminatory power observed by Sperduto et al. This fact can also explain why long-term survival in the poorest prognostic group resembled or even exceeded that of the next group. Again, we cannot exclude the possibility of perfect validation of the GPA in a study with several hundreds of patients. Based on the present survival data (Table 3), it is tempting to propose modification of the poor prognostic group (0–0.5 rather than 0–1 points), and also the best one (3–4 points rather than 3.5–4 points). However, the cohort size of 180 patients does not justify a definitive recommendation regarding these issues. We opted against inclusion of patients treated before 2006, because reduced utilization of brain MRI, PET and sequential lines of chemotherapy in historical patients may diminish the applicability of the validation results.

Rades et al. studied 157 patients treated with WBRT (30 Gy in 10 fractions) [14]. The prognostic factors were identical to those in the new GPA and the present study (age, performance status, number of brain metastases, extracranial metastases). A comparable retrospective database included 221 patients treated with WBRT in Germany [15]. Results were not identical, because prognostic factors included KPS, extracranial disease status and time of appearance of brain metastases (better survival if synchronous). Based on these, a new BMS score was proposed and compared with the well-known Diagnosis-specific graded prognostic assessment (DS-GPA) (Sperduto et al [16]), which is slightly different from the new SCLC GPA. The BMS score was superior.

Previous prognostic models in this setting were not necessarily widely applicable. For example, the SEER database utilized by Shan et al. lacks information about KPS and number of brain metastases [17]. Hou et al. studied four older scores including the DS-GPA in 451 patients treated with WBRT at a single institution (Shanxi Province Cancer Hospital) and proposed a model also called BMS [18]. The independent factors predicting survival in their study included KPS, number of brain metastases, extracranial metastases, and (the only difference to the new GPA) whether treatment had been received before diagnosis of brain metastases. In our study, synchronous presentation/chemotherapy naïve status were not associated with survival. The older scores including DS-GPA and the new BMS all predicted survival. The C-indices of the four groups were 0.55, 0.58, 0.59, and 0.64, respectively.

A unique approach was chosen by the group which has developed the laboratory based LabBM score [6]. They selected patients with SCLC from the Vienna Brain Metastasis Registry and evaluated the prognostic factors, including the blood tests [19]. A total of 489 patients were included. Neurological symptoms were present in 61%. Asymptomatic or oligosymptomatic patients had longer survival (9 versus 5 months, $p = 0.03$) and those with synchronous diagnosis had improved prognosis (9 versus 5 months, $p = 0.008$). Older scores, including DS-GPA ($p < 0.001$), and LabBM ($p < 0.001$) were statistically significantly associated with survival. In multivariate analysis, both DS-GPA, neurological deficits and LabBM score retained statistical significance. None of the other studies discussed here included blood test results, such as anemia, low albumin or high lactate dehydrogenase.

In summary, several groups have already developed three- or four-tiered scores predicting survival in this setting. The differences between these scores in terms of discriminatory power were not striking. Regarding the new GPA developed by Sperduto et al., the question remains: how much progress does it represent? Given that Sperduto et al. have a long track record of refining various GPA scores [1,6,7,16], it is likely that also the new variant

will gain acceptance. The major point to be made after the present validation study is that collapsing the point sum into three or four prognostic strata blurs or deletes important information, namely that patients with 0–0.5 points have very short survival. In the light of newly introduced combined chemo-/immunotherapy approaches for SCLC [20], it will be also important to monitor their impact on brain metastases incidence, patterns of relapse and overall survival [21], e.g., because adjustment of prognostic models and treatment algorithms might become necessary.

**5. Conclusions**

This study supports the prognostic impact of all four parameters contributing to the GPA. The original way of grouping the parameters and breaking the final strata did not give optimal results in this cohort. Therefore, additional validation databases from different countries should be created and evaluated. Survival of patients with 0–0.5 points was very limited, raising questions about the appropriateness of active anti-cancer therapy.

**Author Contributions:** All authors contributed to the study conception and design. Material preparation, data collection and analysis were performed by C.N., I.P. and M.H. The first draft of the manuscript was written by C.N. and A.L.G. and all authors commented on previous versions of the manuscript. All authors have read and agreed to the published version of the manuscript.

**Funding:** This research received no external funding.

**Institutional Review Board Statement:** As a retrospective quality of care analysis, no approval from the Regional Committee for Medical and Health Research Ethics (REK Nord) was necessary. This research project was carried out according to our institutions' guidelines and with permission to access the patients' data.

**Informed Consent Statement:** Informed consent was obtained from all subjects involved in the study. The manuscript does not contain case studies using individual people with identifying information. It is therefore not necessary to provide the form.

**Data Availability Statement:** The dataset supporting the conclusions of this article is available at request from the corresponding author, if intended to be used for meta-analyses.

**Conflicts of Interest:** The authors declare no conflict of interest.

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
