# Peer review of "External Validation of the Graded Prognostic Assessment in Patients with Brain Metastases from Small Cell Lung Cancer"

_curroncol, doi:10.3390/curroncol29100565_

Round 1
Reviewer 1 Report
This is a very nice, well-written study evaluating prognostic factors in a relatively large number of patients with brain metastases from small cell lung cancer (SCLC) treated at two European centers, focused on evaluating the individual components of Sperduto's small cell GPA system as well as the GPA groupings. There is sufficient detail presented clearly and concisely. The discussion is excellent in terms of comparing results with the GPA system, summarizing other literature on prognostic systems for SCLC patients with brain metastases, and discussing limitations of the current study. The conclusions are justified.
Abstract, line 17: Please clarify that survival was measured from diagnosis of brain metastases.
Introduction, line 40: Please clarify what is meant by the term "local treatment." Is this referring to brain metastasis treatment (radiation and/or surgery)?
Introduction, line 49: Again, please add that survival was measured from diagnosis of brain metastases.
Table 2: I whole-heartedly agree with the inclusion of all of the parameters listed in Table 2. I recommend adding a column to provide the hazard ratios that go along with the p-values listed.
Figures: I whole-heartedly agree with providing both Figure 1 and Figure 2.
Discussion, lines 117-130: It would be clearer if all of these comparisons with Sperduto's study maintained the same order: the present study versus Sperduto's study or Sperduto's study versus the present study.
Author Response
Thank you very much for these helpful suggestions, which improve the clarity of the manuscript. All your suggestions have been incorporated in the revised version.
Reviewer 2 Report
Thank you for the opportunity of reviewing this short communication. Even though the topic itself might not be of high impact, this piece stands out with its rigor and scientific soundness. Furthermore, due to the lack of prospective data on treatment choices for brain metastasized SCLC patients, also retrospective data is valuable for decision making in daily practise.
Author Response
Thank you very much for the comments. For changes in the revised manuscript, please see our other response to reviewers.